# Combined HP ^13^C Pyruvate and ^13^C-Glucose Fluxomic as a Potential Marker of Response to Targeted Therapies in YUMM1.7 Melanoma Xenografts

**DOI:** 10.3390/biomedicines10030717

**Published:** 2022-03-19

**Authors:** Chantale Farah, Marie-Aline Neveu, Caner Yelek, Caroline Bouzin, Bernard Gallez, Jean-François Baurain, Lionel Mignion, Bénédicte F. Jordan

**Affiliations:** 1Biomedical Magnetic Resonance Research Group, Louvain Drug Research Institute, Université Catholique de Louvain, UCLouvain, B-1200 Brussels, Belgium; chantale.farah@uclouvain.be (C.F.); caner.yelek@uclouvain.be (C.Y.); bernard.gallez@uclouvain.be (B.G.); 2Laboratory of Tumor Inflammation and Angiogenesis, Department of Oncology, KU Leuven, B-3000 Leuven, Belgium; mariealine.neveu@gmail.com; 3IREC Imaging Platform, Institut de Recherche Expérimentale et Clinique, Université Catholique de Louvain, UCLouvain, B-1200 Brussels, Belgium; caroline.bouzin@uclouvain.be; 4Nuclear and Electron Spin Technologies (NEST) Platform, Louvain Drug Research Institute (LDRI), Université catholique de Louvain (UCLouvain), B-1200 Brussels, Belgium; 5Molecular Imaging and Radiation Oncology (MIRO) Group, Institute de Recherche Expérimentale et Clinique (IREC), B-1200 Brussels, Belgium; jean-francois.baurain@uclouvain.be

**Keywords:** melanoma, tumor metabolism, targeted therapy, BRAF and MEK inhibition, ^13^C-MRS, markers of response

## Abstract

A vast majority of BRAF V600E mutated melanoma patients will develop resistance to combined BRAF/MEK inhibition after initial clinical response. Resistance to targeted therapy is described to be accompanied by specific metabolic changes in melanoma. The aim of this work was to evaluate metabolic imaging using ^13^C-MRS (Magnetic Resonance Spectroscopy) as a marker of response to BRAF/MEK inhibition in a syngeneic melanoma model. Tumor growth was significantly delayed in mice bearing YUMM1.7 melanoma xenografts treated with the BRAF inhibitor vemurafenib, and/or with the MEK inhibitor trametinib, in comparison with the control group. ^13^C-MRS was performed in vivo after injection of hyperpolarized (HP) ^13^C-pyruvate, at baseline and 24 h after treatment, to evaluate dynamic changes in pyruvate-lactate exchange. Furthermore, ex vivo ^13^C-MRS steady state metabolic tracing experiments were performed after U-^13^C-glucose or 5-^13^C-glutamine injection, 24 h after treatment. The HP ^13^C-lactate-to-pyruvate ratio was not modified in response to BRAF/MEK inhibition, whereas the production of ^13^C-lactate from ^13^C-glucose was significantly reduced 24 h after treatment with vemurafenib, trametinib, or with the combined inhibitors. Conversely, ^13^C-glutamine metabolism was not modified in response to BRAF/MEK inhibition. In conclusion, we identified ^13^C-glucose fluxomic as a potential marker of response to BRAF/MEK inhibition in YUMM1.7 melanoma xenografts.

## 1. Introduction

Melanoma is considered the most devastating form of skin cancer with an increasing incidence over the last decades [1]. Early detected melanoma are highly curable, but the metastatic forms are highly refractory to treatments, with a 5-year survival for only 25% of diagnosed patients [1]. Strikingly, 50% of melanomas harbor BRAF V600E mutation that consists in the substitution of valine for glutamic acid at codon 600, that results in the constitutive activation of the serine/threonine kinase activity of BRAF. BRAF is part of the MAPK (Mitogen Activated Protein Kinase) signal transduction pathway along with the downstream effectors MEK and ERK, playing a crucial role in cell proliferation, differentiation and apoptosis [2,3,4].

Since the 70′s and the approval of the alkylating agent dacarbazine [4], the treatment of melanomas has significantly evolved, including targeted therapies and immunotherapies. Targeted therapies based on BRAF or MEK inhibition have immediate effects. Indeed, clinical trials have shown a significant increase in survival time in patients treated with vemurafenib (BRAF inhibitor) versus dacarbazine [5]. However, targeted therapies show only short-term benefits due to the development of resistance. Acquired resistance are primarily associated with MAPK pathway reactivation driven by mutations in NRAS and MEK1 [6,7,8]. In the purpose, combined BRAF and MEK inhibition have significantly improved overall survival in melanoma patients [9,10]. However, resistance to the treatment combination also occurs. Interestingly, immunotherapies based on immune checkpoint inhibitors do offer a long-term effect with durable benefit although they may show lower response rate than targeted therapies in melanoma [9,10,11,12]. In BRAF-mutated melanoma, the response rate to combined BRAF and MEK inhibitors is around 70% [1]. With immune checkpoint inhibitors (ICI), a 58% response rate to anti-PD1 was reported [1].

MAPK signaling, tumor metabolism and resistance to BRAF inhibitors are suggested to be interconnected in melanoma [13]. Within the scope, melanoma cancer cells have been described to exhibit aerobic glycolysis [14]. Indeed, in the context of MAPK signaling, mutant BRAF promotes glycolytic activity and inhibits oxidative phosphorylation (OXPHOS) via negative regulation of the MITF- PGC1α axis [15,16]. The BRAF inhibitor vemurafenib, is suggested to reduce glycolytic metabolism and to activate mitochondrial metabolism via several mechanisms [17], including inhibition of hexokinase II (HKII) and glucose transporter I and III (GLUTI/III) expression [18]. This is corroborated by the work of Haq et al., showing that vemurafenib induces significant increase in the expression of mediators involved in OXPHOS as well as in citric acid cycle, promoting mitochondrial biogenesis and oxidative metabolism [16]. In addition to glycolysis, glutamine dependency has been observed in melanoma, with a switch from glucose to glutamine metabolism upon resistance to therapy [14,19,20,21]. In the purpose, Scott et al. have observed an increased glutamine metabolism in melanoma cells relying on the Warburg effect and indicated that glutamine is an essential nutrient for melanoma cells as much as glucose [14].

The identification of robust response biomarkers remains a challenge to assess drug resistance [22]. Several biomarkers have been studied in metastatic melanoma such as clinical biomarkers (tumor burden and metastatic sites), blood markers (serum LDH, neutrophils, monocytes and lymphocytes levels), stool (gut microbiome), tissue markers (mutational analysis, tumor infiltrating lymphocytes) and imaging biomarkers [23]. Among several non-invasive molecular imaging techniques currently used in the preclinical and clinical settings, ^18^F-flouro-deoxy-glucose-positron emission tomography (FDG-PET) was suggested to be useful as a marker of melanoma response [24]. Indeed, a decrease in FDG uptake was shown in tumor cells in vitro and in tumors in vivo after drug treatment [25]. Specifically, BRAF inhibition has led to low FDG uptake in BRAF-mutated melanomas in xenografts as well as in patients [26,27].

^13^C-metabolic imaging is now considered to assess sensitivity to therapy in the preclinical setting as well as in clinical studies, with a higher specificity than ^18^F-FDG, by monitoring the fate of glucose beyond glucose uptake by the tumor cells [28,29]. The monitoring of ^13^C enriched metabolic substrates can be monitored using tools such as ^13^C magnetic resonance spectroscopy (MRS) or imaging (MRI). MRS can investigate static metabolic processes in vivo after injection of a ^13^C enriched substrate (i.e., ^13^C-glucose, ^13^C-glutamine) [30]. Moreover, dynamic metabolic processes can be assessed by temporarily boosting the ^13^C NMR signal of some key metabolic substrates (such as ^13^C-pyruvate), using hyperpolarization, thereby allowing dynamic measurement of metabolic conversions in vivo [31]. Notably, Dynamic Nuclear Polarization (DNP) can increase ^13^C-MRS sensitivity by 10,000-fold and provides an opportunity to detect real time metabolic fluxes, such as ^13^C-pyruvate ^13^C-lactate label exchange [32].

Following treatment with BRAF inhibitors [16,17] or MEK inhibitors [33], mutant BRAF human cancer cells (WM266.4 and SKMEL28) showed an inhibition of hyperpolarized (HP) ^13^C-pyruvate-lactate exchange, associated with depletion in hexokinase 2 and monocarboxylate transporters (MCT) 1 and 4 [17]. Similarly, BRAF inhibitors in human A375 sensitive melanomas cells impaired glycolysis in vitro, as attested by a decreased ^13^C-pyruvate -^13^C-lactate exchange in response to vemurafenib [34]. However, in human A375 melanoma xenografts in-vivo, the HP pyruvate-lactate exchange was increased after BRAF inhibition [34]. This paradoxical effect suggested a significant influence of the tumor microenvironment on the tumor metabolic phenotype [34].

In this study we further characterized the ^13^C-metabolic profile in response to BRAF/MEK targeted therapies in YUMM1.7 syngeneic melanoma xenografts characterized for BrafV600E/wt Pten−/− Cdkn2−/−, allowing the use of immunocompetent mice to fully integrate all aspects of the tumor microenvironment. We aimed to evaluate the relevance of hyperpolarized (HP) ^13^C -pyruvate as well as of ^13^C-MRS fluxomic after [U-^13^C]-glucose or [5-^13^C]-glutamine injection, as potential markers of response to targeted therapies in the preclinical setting.

## 2. Materials and Methods

### 2.1. Tumor Models

Experiments involving animals were undertaken in accordance with the Belgian law concerning the protection and welfare of the animals and were approved by the UCLouvain ethical committee (agreement reference: UCL/2018/MD/021). Animals were housed in animal facility under standard conditions of temperature 20–24 °C and humidity between 45–65%. All investigators performing in vivo studies successfully completed FELASA C training.

YUMM1.7 mouse malignant melanoma cell line was purchased from American Type Cell Culture (ATCC, Manassas, VA, USA) and cultured in Dulbecco’s Modified Eagle Medium (DMEM) supplemented with 10% heat inactivated fetal bovine serum (GIBCO, Thermo Fisher Scientific, Waltham, MA, USA).

Cells were harvested by trypsinization and resuspended in PBS (pH 7.4) before injection to animals. 10^6^ YUMM1.7 cells in 50 µL PBS were intradermically injected in the right hind paw of specific pathogen-free (SPF) 8 weeks-old female C57BL-6 mice (Janvier Labs). During inoculation, mice were kept under inhalational anesthesia with isoflurane 2.5% in 2 L/min airflow.

### 2.2. Animal Treatment

Following tumor inoculation, when the xenografts reached 300 mm^3^+/−50 mm^3^, mice were randomized into 4 groups and treated via daily intraperitoneal injections of the BRAF inhibitor PLX-4032 (25 mg/kg), MEK inhibitor GSK-112021 (0.5 mg/kg), or the combination of BRAF inhibitor and MEK inhibitor, or vehicle (30% DMSO in 120 µL PBS). PLX-4032 (vemurafenib) and GSK112021 (trametinib) were purchased from Bioconnect. After 5 doses of daily treatment, treatments were interrupted, and tumor regrowth was longitudinally monitored using an electronic caliper. The growth delay of melanoma xenografts was calculated as the time, in days, to reach the volume of 800 mm^3^.

### 2.3. Tissue Fixation and Freezing

Mice were killed by cervical dislocation, the individual tumor (800 mm^3^) from each study animal was collected and divided in half. One half of the tumor was immediately snap frozen in liquid nitrogen and stored at 80 °C until protein extraction. The remaining portion was immediately fixed in 4% paraformaldehyde for 24 h at room temperature. Samples were subsequently transferred into an automated tissue processor and embedded in paraffin. Following deparaffinization, inactivation of endogenous peroxidases, antigen retrieval in citrate buffer and non-specific binding blocking, 5 µm sections were incubated overnight at 4 °C with the primary antibodies for p-ERK (Cell Signaling Technology, ref. #4370S, 1:100 dilution), MCT1 (Proteintech, 20139-1-AP, 1:1000 dilution) and MCT4 (Sigma, HPA021451, 1:500 dilution) Consequently, sections were incubated at room temperature for 30 min with Envision antirabbit secondary antibody (Dako, ref. #K4003) and stained with diaminobenzidine for 5 min (Dako, #K3468). Stained slides were then digitalized using a SCN400 slide scanner (Leica Biosystems, Wetzlar, Germany) at X20 magnification and analyzed using Visiopharm Software. The quantification algorithm was run in the viable part of the tissue samples to detect stained area and analyzed tumor area. A % of stained area was calculated as the ratio between the stained area and the analyzed tumor area multiplied by 100.

### 2.4. Homogenisation of Tumors

Tumor samples were grinded using a pestle at cryogenic temperature. Following grinding, the fragmented pieces were transferred into a tube and stored at −80° for further analysis (Western Blot, ^13^C-MRS experiments)

### 2.5. Western Blot

Homogeneous tumor powder was lysed in RIPA buffer (Thermo Scientific, Waltham, MA, USA) supplemented with 1% protease and phosphatase inhibitors (Thermo Scientific). Protein amount was measured with a Pierce^TM^ BCA protein Assay Kit (Thermo Scientific). Equal amounts of proteins were loaded onto 4–15% Mini-PROTEAN TGX^TM^ Precast Gels (Bio-Rad). Following electrophoresis in 1× Tris/glycine/SDS running Buffer (Bio-Rad), proteins were transferred to PVDF membranes using the Trans-Blot Turbo RTA Mini PVDF Transfer Kit (Bio-Rad) according to the vendor’s instructions. Non-specific binding was blocked by soaking the membranes in 5% BSA in tTBS (1* Tris-Buffered Saline, 0.1% Tween 20, Bio-Rad) at room temperature for 1 h.

Membranes were incubated with primary anti-HSP90, anti-LDHA (Cell Signaling, #2021S, dilution 1:1000), in tTBS-BSA 5% at 4 °C overnight, followed by incubation with anti-rabbit or anti-mouse secondary antibodies (Jackson IR) in tTBS-BSA 1% at room temperature for 1 h. Detection was performed using the SuperSignal^TM^ West Pico Plus Kit (Thermo Scientific) and an ImageQuant LAS 500 camera (GE Healthcare). Quantification was performed on ImageJ by measuring the integral of the optical density profile of the band of the expected molecular weight. No Background correction was performed.

### 2.6. Hyperpolarized ^13^C-MRS

Hyperpolarized ^13^C-NMR data were acquired as previously described [34].

For hyperpolarization experiments, 40 µL of [1-^13^C] pyruvic acid (Sigma-Aldrich, Saint Louis, MO, USA) solution containing 15 mmol/L of trityl radical OXO63 and 2 mmol/L gadolinium were hyperpolarized in an Oxford Dynamic Nuclear Polarizer (HyperSense, Oxford, UK) for approximately 45 min at 1.4 K and 3.35 T. The polarized solution was rapidly dissolved in 3 mL of a heated buffer containing 100 mg/L EDTA, 40 mmol/L HEPES, 30 mmol/L NaCl, 80 mmol/L NaOH, 30 mmol/L of non-HP unlabeled lactate. This solution was quickly injected using a catheter into the tail vein of the mice in the MRI scanner (11.7-Tesla, Bruker, Biospec, NEST Platform, UCLouvain). Mice were scanned using a double tuned ^1^H-^13^C-surface coil (RAPID Biomedical, Rimpar, Germany) as previously described, which was designed for spectroscopy of subcutaneous tumors.

In this case, ^13^C spectra acquisition and infusion of HP [1-^13^C] pyruvate were started simultaneously. During MR experiments, animals were kept under inhalational anesthesia with isoflurane (2.5% during anesthesia induction, 1–2% during maintenance) in 2 L/min airflow. Temperature was continuously monitored and kept at 37 °C ± 1 °C via a warmed blanket. Spectra were acquired at 37 °C every 3 s for 210 s. ^13^C label exchange between HP [1-^13^C] pyruvate and [1-^13^C] lactate was measured as the ratio between the corresponding areas under the curve (AUC) via a homebuilt Matlab routine (The MathWorks Inc, Portola Valley, CA, USA).

### 2.7. ^13^C-MRS

For in vivo ^13^C-glucose and ^13^C-glutamine fluxomic experiments, ^13^C-glucose (2 g/Kg in PBS) at 15 min intervals (t0–15–30 min) 3 times) or ^13^C glutamine (800 mg/Kg) were injected intravenously at day 1 into the tail vein of tumor bearing mice (according to the protocol published by Yuan and colleagues) [35]. Tumors were resected 24 h after treatment initiation and Ex vivo ^13^C-MRS steady state metabolic profiling has been performed as described previously [36] and analyzed on a high-resolution 600 MHz NMR (Bruker Ascend, NEST platform, UCLouvain). The acquisition time was 0.8 s with 2048 repetitions and 10 s of interpulse delay (1D sequence with inverse gated decoupling using 30° flip angle). Spectrum analysis and quantification were performed with MestReNova software version 14.2.0-26256 (Santiago de Compostela, Spain). Metabolites were quantified by peak integration relative to internal standards and corrected for tumor mass per sample.

### 2.8. Statistical Analysis

One- and two-way anova analysis were performed via Graphpad Prism9.1.2 (software) followed by Holm-Sidak multiple comparisons test, with *p <* 0.05 considered significant. Results are represented as mean ± SEM. 

## 3. Results

### 3.1. Combined BRAF and MEK Inhibition Delays Tumor Growth to a Larger Extent Than BRAF or MEK Inhibition Alone in Syngeneic YUMM1.7 Melanoma Xenografts

To assess the efficacy of BRAF and MEK targeted therapies in YUMM1.7 xenografts, C57Bl6 mice bearing YUMM1.7 tumors reaching 300 mm^3^+/−50 mm^3^ were treated with daily intraperitoneal injection for 5 days of the BRAF inhibitor (BRAFi) vemurafenib (25 mg/kg), the MEK inhibitor (MEKi) trametinib (0.5 mg/kg) or of a combination of both vemurafenib and trametinib (BRAFi+MEKi: combo), or with vehicle (DMSO 2.5%) (Figure 1A). Both single BRAF_i_, MEK_i_ as well as BRAF_i_/MEK_i_ combination delayed the growth of YUMM1.7 melanoma xenografts, with a significantly longer delay being induced by the combination (relative growth delay factor (RGD) = 1.97, *p =* 0.04 in BRAF_i_-treated mice mice and RGD = 3, *p =* 0.0005 in MEK_i_ treated mice and RGD = 4.95, *p <* 0.0001 in BRAF_i_/MEK_i_ treated) (Figure 1B).

To confirm the effective inhibition of the target, we performed immunohistochemistry for phosphorylated ERK (p-ERK) on YUMM1.7 melanoma xenografts collected 4 h after single treatment with BRAF_i,_ MEK_i,_ or combined BRAF_i_/MEK_i_. P-ERK levels were reduced in all groups, reaching significance in response to MEK inhibition at this time point (*p =* 0.03), in the BRAF_i_ treated group (*p =* 0.1) and for the combination (*p =* 0.07)(Figure 1C). Representative examples for p-ERK staining of melanomas collected 4 h after treatment are shown on Figure 1D. We also performed immunohistochemistry for the proliferation marker Ki-67 in YUMM1.7 melanoma xenografts collected 24 h after single treatment with BRAF_i,_ MEK_i,_ or combined BRAF_i_/MEK_i_ (combo). Ki-67 levels showed a decrease, although not statistically significant, at 24 h in all treated groups. (Figure 1E). Representative examples for staining of Ki-67 melanomas collected 24 h after treatment are shown on Figure 1F.

### 3.2. Hyperpolarized ^13^C Pyruvate -Lactate Exchange Is not Modified in Response to BRAF and/or MEK Inhibition

Following intravenous injection of HP [1-^13^C] pyruvate to mice bearing YUMM1.7 xenografts, we monitored the dynamic evolution of the ^13^C signal from [1-^13^C] lactate and [1-^13^C] alanine (Figure 2A) using in vivo ^13^C-MRS (Bruker Biospec, 11.7T). Figure 2B shows representative evolutions of ^13^C pyruvate and ^13^C lactate peaks over time, from which the area under the curve and ratios were calculated. The ^13^C label exchange between HP pyruvate and lactate, represented by the lactate to pyruvate ratio, decreased in 4 out of 5 tumors 24 h after a single dose of treatment in the combination group (Figure 2C). However, the ratio was not significantly modified while comparing means of all groups. (Figure 2D). Of note, ^13^C signal arising from alanine was very low and showed no significant change at baseline and 24 h after a single dose of treatment.

### 3.3. In Vivo ^13^C Glucose Fluxomic, and Not ^13^C Glutamine, Detects Metabolic Changes in Response to BRAF and/or MEK Inhibition

In order to further characterize the metabolic profile of YUMM1.7 xenografts treated with BRAF/MEK inhibitors, fluxomic experiments were performed after injection of either uniformly labeled ^13^C-glucose or 5-^13^C-glutamine. The downstream metabolites were detected after tumor resection, performed at 24 h after a single dose of treatment, for metabolic profiling using high-resolution ^13^C-NMR (Bruker Ascend, 600 MHz).

Typical spectra acquired before and after treatment combination (BRAFi/MEKi: combo) are shown on Figure 3A. A significant decrease was observed in ^13^C-lactate production in response to single BRAF inhibition (*p =* 0.003), to single MEK inhibition (*p =* 0.003), as well as in response to the combined BRAF and MEK inhibitions (*p =* 0.003), in comparison with the control group, at 24 h post treatment. (Figure 3B). This reduction suggests a decrease of the glycolytic metabolism in the YUMM1.7 melanoma xenografts in response to single BRAF or MEK inhibition, as well as in response to the combined BRAF and MEK inhibition. However, the technique did not evidence any additional effect of the combined inhibition with respect to the single BRAF or MEK inhibitions. Of note, no significant change was observed in the production of ^13^C-alanine.

Regarding the metabolites issued from 5-^13^C glutamine, we did not observe any significant change in ^13^C-glutamate production. (Figure 3C,D).

### 3.4. The Expression of Metabolic Transporters or LDH-A Are Not Modified in Response to BRAF or MEK Inhibition in YUMM1.7 Xenografts

In order to characterize the potential decreased glycolytic activity observed in tumors treated with the combined BRAF/MEK inhibitors, we assessed expression of GLUT-1 and of the monocarboxylate transporters MCT1 and 4, as well as the LDH-A activity at 24 h post-treatment. No important modification was observed, in line with the lack of change in the HP lactate to pyruvate ratio, which is described to be dependent on multiple factors including monocarboxylate transporters and/or LDH-A activity depending on the cell line. (Figure 4A–H). Of note, MCT1 staining was surprisingly increased in response to MEK inhibition only, yet with a large standard deviation in this group, which remains so far unexplained.

## 4. Discussion

BRAF and MEK inhibitors have shown clinical benefit in patients with BRAF-mutant melanoma [37,38]. However, if the BRAFi/MEKi combinations have improved the outcomes and overall survival, resistance still occurs [39]. Biomarkers are needed to identify patients that respond or resist to treatment. Therefore, our aim was to evaluate the relevance of hyperpolarized (HP) ^13^C -pyruvate and ^13^C-MRS fluxomic after U-^13^C-glucose or 5-^13^C-glutamine injection as markers of response to targeted therapies in syngeneic melanoma xenografts. In the syngeneic YUMM1.7 model, we observed that the combination of BRAF and MEK inhibition delayed the tumor growth more significantly than the single agents, in accordance with recent studies [40]. The p-ERK levels, assessed at 4 h post-treatment, showed a trend to a decrease in response to all treatments, in accordance with the efficacy of the targeted treatments. However, pERK reduction reached statistical significance in response to MEK inhibition only, showing that this molecular marker is not robust enough to predict response on an individual basis, although additional time points should be explored. Proliferation assessed at 24 h post treatment using ki67 staining also showed a trend to a decrease in all groups but never reached significance, suggesting that additional mechanisms account for tumor growth delay in response to BRAF/MEK inhibition in YUMM1.7 xenografts. It was indeed described that vemurafenib induces senescence and apoptosis in melanoma cell lines via mechanism involving caspases 3, ref. [41] while apoptosis induction by MEK inhibition is described to be caspases-independent [42]. Further experiments to assess apoptosis in YUMM1.7 melanoma xenografts are required to better characterize the effect of MAPK inhibitors in YUMM1.7 xenografts.

Dynamic nuclear polarization has recently entered the clinical setting, allowing the study of metabolic processes in real time using hyperpolarized ^13^C pyruvate. It was first reported in prostate cancer patients [28], and later implemented in several clinical trials in breast, brain and cervical cancer [43]. Furthermore, ^13^C-glucose feeding fluxomic experiments have been used to understand metabolic processes in human tumors, including brain and pediatric tumors [44,45].

Within the scope, we have assessed HP ^13^C-pyruvate-lactate exchange 24 h after treatment with BRAF/MEK inhibitors in YUMM1.7 xenografts. We observed a lack of change in the lactate to pyruvate ratio assessed at 24 h post-treatment. Of note, the HP pyruvate to lactate ratio was shown to be significantly increased following single BRAF inhibition in immunodeficient mice bearing A375 human melanoma xenografts [34]. These differences observed between a syngeneic and a human model could reflect the effect of the contribution of the tumor microenvironment on the metabolic interactions, suggesting that such conversions are influenced by the microenvironment, potentially including immune cells.

It has been described in a recent study that MCT1, the bidirectional transporter for monocarboxylic acids (such as lactate or pyruvate) correlates with glycolytic metabolism and malignancy, its inhibition has been described to impact the HP ^13^C-lactate to ^13^C- pyruvate ratio. Within the scope, the hyperpolarized ^13^C pyruvate to lactate conversion has been shown to be rate limited by the monocarboxylate transporter in the plasma membrane in some models [46]. Overall, the pyruvate to lactate ratio is dependent on multiple factors including the monocarboxylate transporters MCT1, MCT4 but also LDHA expression and activity, depending on the tumor models. Therefore, we performed ex vivo immunohistochemistry analysis of MCT1, MCT4 and LDH-A. We did not observe any significant effect of BRAF and/or MEK inhibition, in accordance with the HP pyruvate to lactate exchange data, except for the MCT1 expression that was surprisingly increased in the MEKi treated group, yet with a huge standard deviation in this group. This should be further explored on a larger sample size.

Next, to better understand the metabolic changes in response to BRAF and MEK inhibition we performed an NMR metabolic profiling using uniformly ^13^C labeled glucose or ^13^C_5_ glutamine. Contrarily to what we observed with HP ^13^C-MRI, we found a significant decrease in the ^13^C lactate production after ^13^C-glucose injection. The lack of change in GLUT-1 expression in response to BRAF/MEK inhibition in YUMM1.7 xenografts suggest that this reduction in not due to a decrease in glucose uptake by the cells, but potentially to a decrease in the glycolytic activity of the tumor cells. This metabolic shift was also shown in vitro by the group of Beloueche-Babari in WM266.4 cells in response to single BRAF inhibition [47]. The decrease in ^13^C lactate production in vemurafenib and trametinib treated groups, compared to control, was in line with studies suggesting that vemurafenib reduces glycolytic metabolism and activates mitochondrial metabolism [16,22,47,48]. Of note, we did not observe any significant change in the second quantified ex vivo metabolite, ^13^C alanine, which is another product of glucose metabolism. If changed it would suggest a major role of aminotransferase enzyme (ALT) in the generation of alanine from pyruvate. Altogether, these results suggest that the melanoma xenografts treated with BRAF and MEK inhibitors produce less lactate and are likely to be less glycolytic while responding to treatment A hypothesis is that this reduction in lactate production could be related to a reduced hexokinase (HK-2) activity, and/or to a modification of the activity of other key glycolytic enzymes such as pyruvate kinase dehydrogenase (PDK). However, the lack of differences between the combination group with respect to the single inhibitors suggest that more sensitive metabolic makers are yet to be identified. In addition, the ^13^C glutamine tracing experiments did not show any effect of BRAF/MEK inhibition on glutamate production, suggesting that this model is not particularly relying on the glutamine pathway. Accordingly, evidence from the few studies that have infused ^13^C-glutamine into mouse models of cancer reported a complex context specificity for the use of this fuel [47]. Moreover, glutamine dependency is mostly described in BRAF/MEK inhibitors resistant models. Further studies are therefore required in resistant models in the scope of ^13^C-glutamine fluxomic experiments.

Taken together, these results suggest that ^13^C glucose fluxomic can identify metabolic changes in response to BRAF/MEK inhibition in vivo in a syngeneic model and that ^13^C-glucose is a specific biomarker of response to detect treated and untreated tumors. Furthermore, HP ^13^C pyruvate could discriminate positive response in 4 out of 5 tumors treated with the combined BRAF/MEK inhibition (combo), in comparison with the single inhibitors, showing potential for a better sensitivity when used for individual monitoring. In conclusion, the combined assessment of metabolic changes using HP ^13^C pyruvate and ^13^C -glucose fluxomic could constitute an ideal multi-modal approach to assess response to BRAF/MEK inhibition in melanoma with specificity and sensitivity. This approach should be further assessed as a marker of response to targeted therapy in melanoma, considering their high translational potential into the clinical setting. Further studies are yet needed to better understand the metabolic response of melanoma to BRAF/MEK inhibition in the YUMM1.7 model.

## Figures and Tables

**Figure 1 biomedicines-10-00717-f001:**
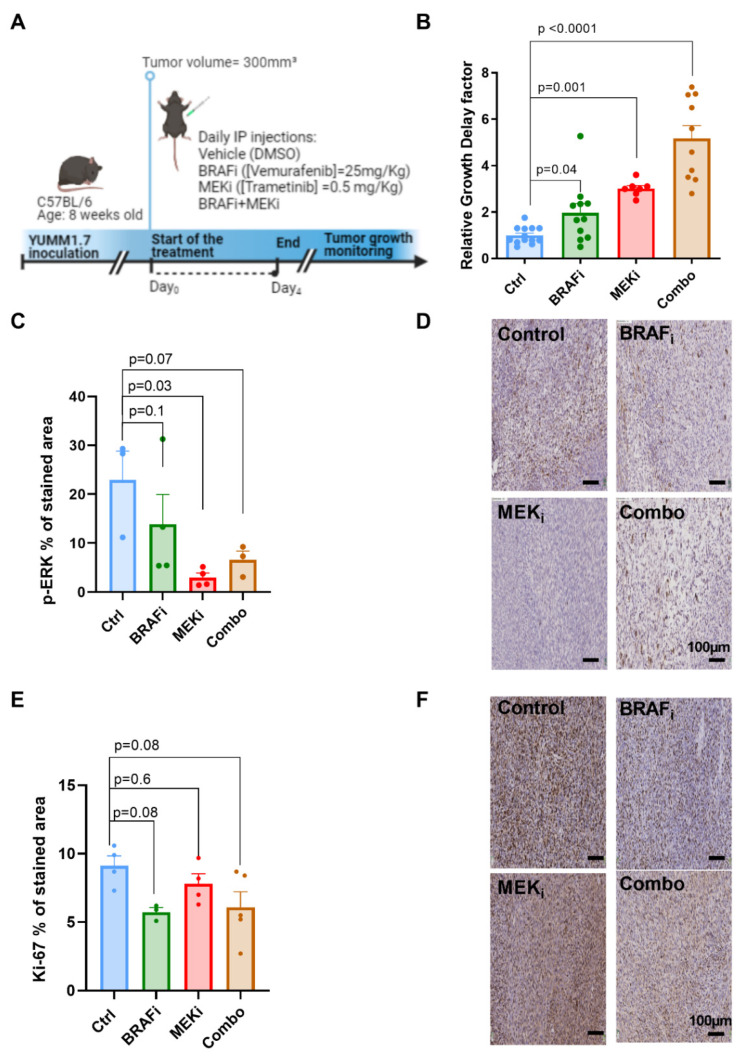
Growth delays, p-ERK and Ki-67 staining in YUMM1.7 melanoma xenografts. Timeline of the in vivo protocol (**A**). The tumor growth delay of melanoma xenografts was calculated as the time, in days, to reach the volume of 800 mm^3^; sample size *n* = 13 for Ctrl, *n* = 11 for BRAF_i_ and *n* = 7 for MEK_i_ and *n* = 10 for Combo (**B**). p-ERK % of stained area at 4 h after treatment and Ki-67 % of stained area at 24 h after treatment. The % of stained area was calculated as the ratio between stained and tissue are multiplied by 100. (**C**,**D**). p-ERK and Ki-67 representative staining of melanoma xenografts collected 4 h and 24 h, respectively, after a single injection of the indicated treatments. (**D**–**F**) All scale bars = 100 µm.

**Figure 2 biomedicines-10-00717-f002:**
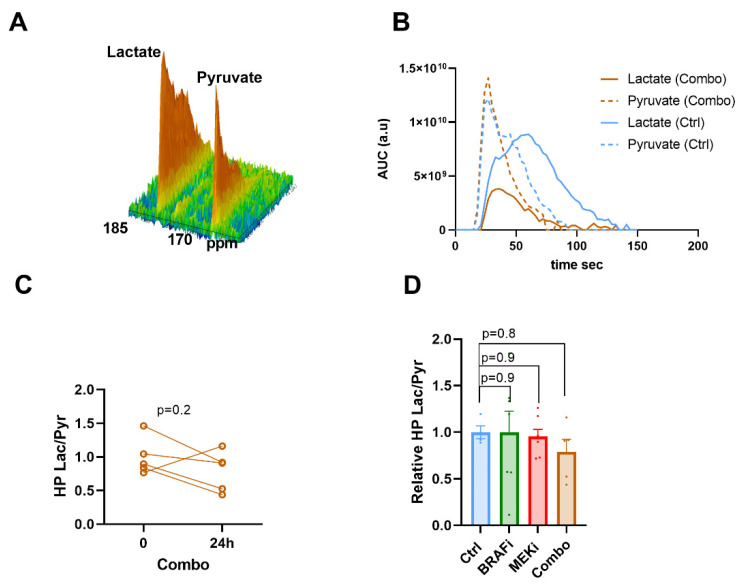
In vivo monitoring of hyperpolarized ^13^C pyruvate -lactate exchange in response to BRAF and MEK inhibition. Representative spectra of the ^13^C signal time course, obtained from a mouse at baseline (**A**). Representative evolution of the area under the curve of ^13^C pyruvate and ^13^C lactate peaks over the time with and without treatment (**B**). Individual changes of calculated lactate/pyruvate ratio’s in the combo group (**C**). ^13^C label exchange between HP pyruvate and lactate (measured as the ratio AUC of [1-^13^C] lactate/AUC of [1-^13^C] pyruvate) in melanoma xenografts 24 h after indicated treatments and normalized to Ctrl. (**D**) One-way anova, Holm-Sidak multiple comparisons test, *n* = 4 for Ctrl, *n* = 7 for BRAF_i_ and MEK_i_, *n* = 5 for BRAF_i_/MEK_i_(combo).

**Figure 3 biomedicines-10-00717-f003:**
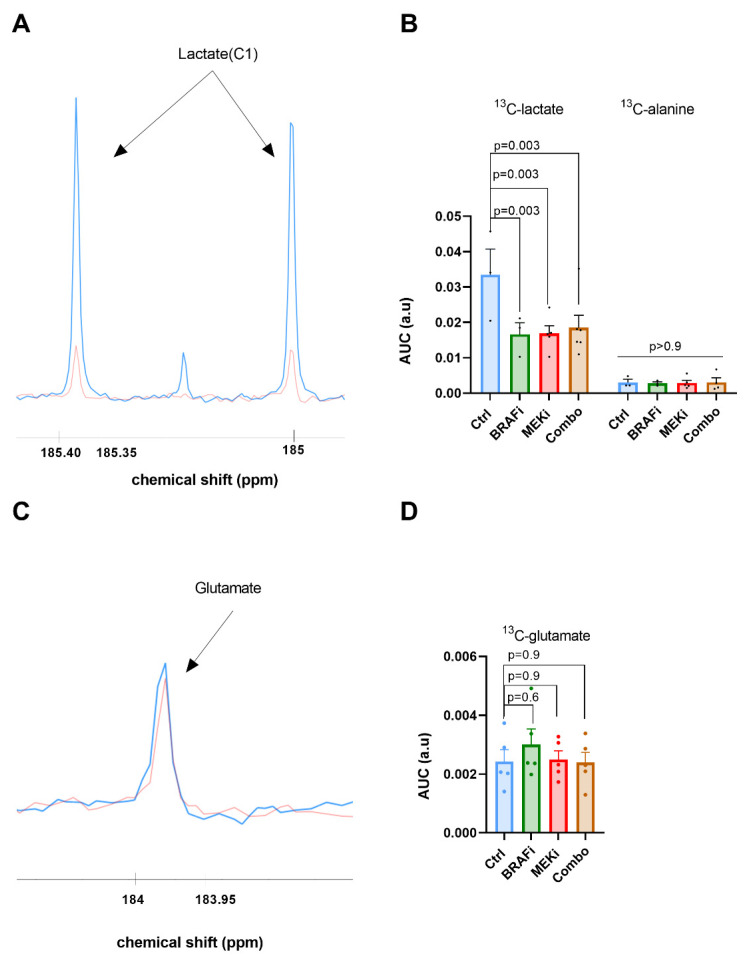
Ex vivo ^13^C metabolic profiling after ^13^C glucose or ^13^C glutamine feeding (fluxomic experiments) in response to BRAF and/or MEK inhibition. Representative spectra of lactate C1 issued from the ^13^C glucose metabolism in the control group (blue) and in the BRAFi/MEKi treated group (orange) (**A**). Quantification of the ^13^C detectable metabolites in YUMM1.7 xenografts, represented by the area under the curve corrected for internal standard (TSP) and tumor mass, arbitrary units (a.u.), two-way anova, Holm-Sidak multiple comparisons test. Sample size *n* = 3 for Ctrl and BRAF_i_ and *n* = 5 for MEK_i_ and BRAFi/MEK_i_ (Combo) (**B**). Representative spectra of glutamate issued from the ^13^C glutamine metabolism in the control group (blue) and in the BRAFi/MEKi treated group (orange) (**C**), and ^13^C glutamate quantification (a.u.), one-way anova, Holm-Sidak multiple comparisons test *n* = 5/grp (**D**).

**Figure 4 biomedicines-10-00717-f004:**
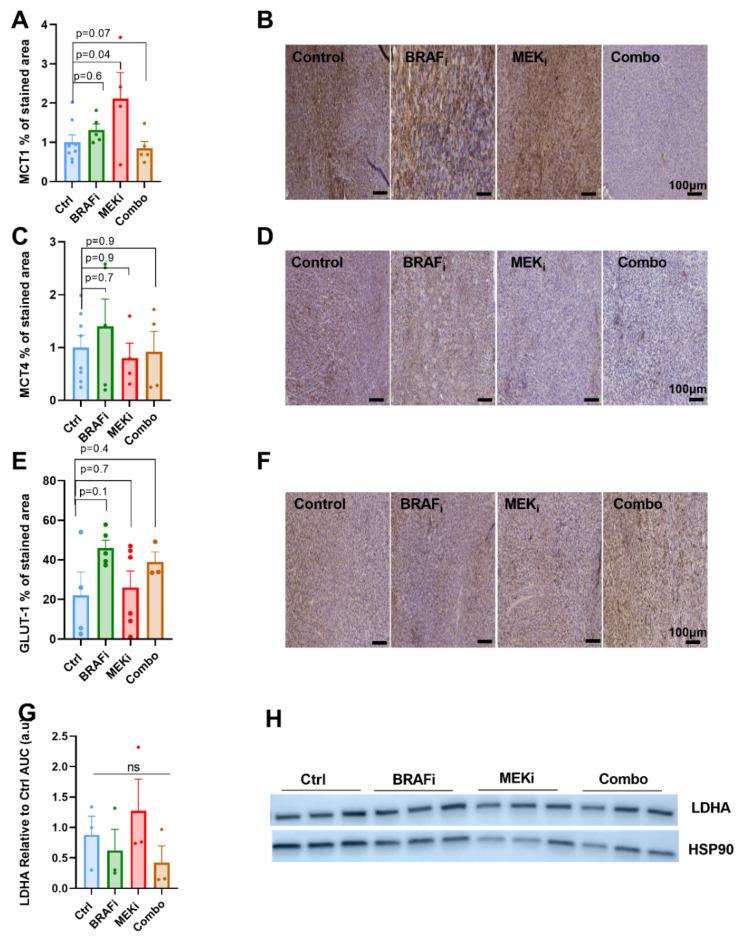
Expression of metabolic transporters and LDH-A in response to BRAF or MEK inhibition in YUMM1.7 xenografts. MCT1, MCT4 and GLUT-1 staining area (relative to control) from xenografts obtained 24 h after single injection of the indicated treatments All scale bars = 100 µm. One-way anova, Holm-Sidak multiple comparisons test. Sample size: *n* = 8 for Ctrl and *n* = 5 for BRAF_i_, MEK_i_ and Combo group (**A**–**F**). AUC analysis of LDHA. Corresponding to the (**G**) Western blot of the lactate deshydrogenase LDHA 24 h after treatments (**H**).

## Data Availability

The data presented in this study are available on request from the corresponding author.

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
