# Peer review of "Combined HP 13C Pyruvate and 13C-Glucose Fluxomic as a Potential Marker of Response to Targeted Therapies in YUMM1.7 Melanoma Xenografts"

_biomedicines, 2022, doi:10.3390/biomedicines10030717_

Round 1

Reviewer 1 Report

General:

It is clearly an important topic to monitor in vivo the response to treatment. The authors follow here an elegant approach by the use of hyperpolarized substrates. At the moment the results are not fully conclusive, and it is not fully clear yet how they might apply to real human patients. Still this paper merits publication. The paper is written in clear and concise English

Specific points:

When  looking at Figure 2B it seems as especially at around 50 seconds in the combo group more lactate is observed than in controls, while pyruvate seems fairly similar in both groups. This would hint at an increased lactate to pyruvate ratio instead of decreased ratio as shown in Figure 2D. Please comment.

Fig 3B, 3D AUCs relative to controls are given. This implies that AUCs of controls should be set to 1 for both lactate and alanine. Please check figures.

Author Response

We would like to thank you for your report, and for noticing these points.

In fact, the figure 2B was supposed to be a typical spectrum of a sample belonging to combo group. Here, it seems we have chosen the only sample where the lactate increased (as pointed in fig 2C and in the discussion of the paper, 4 out of 5 tumors show a decrease in the lac/pyr ratio, but one does show an increase). For the sake of clarity and consistency, we now replaced it with a typical spectrum that belongs to the group of tumors showing a decrease in the lac/pyr ratio.

Thank you for noticing the mistake in Fig.3B as well. In fact, the AUC is normalized relative to an internal standard, TSP (as explained in the legend) and not to the control group. We modified the Y axis legend accordingly.

Reviewer 2 Report

The manuscript submitted by Farah et al. explored the impact of BRAF/MEK targeted therapies in YUMM1.7 syngeneic melanoma xenographs by measuring the 13C-metabolic profile through MRS.

It is:

  • studied the influence of BRAF and MEK inhibition (BRAFi and MEKi) , alone or combined; on tumor growth, in hyperpolarized 13C lactate to pyruvate ratio; on expression of LDH-A and metabolic transporters and on 13C glucose and 13C glutamine metabolism.
  • shown that BRAFi and/or MEKi delays tumor growth, reduces p-ERK; and cell proliferation.
  • shown that lactate, but not glutamate, is markedly reduced by BRAFi/MEKi group
  • shown that expression of LDH-A, monocarboxylate transporters 1 and 4 and GLUT-1 is not altered, except MCT-1 expression under MEKi.

It is concluded that “the combined assessment of metabolic changes using HP 13C pyruvate and 13C-glucose fluxomic could constitute an ideal multi-modal approach to assess response to BRAF/MEK inhibition in melanoma with specificity and sensitivity”.

The topic is relevant, the study is well designed and the methods appropriate. The results are properly discussed, except for changes in MCT1 expression caused by MEKi, which should be done.

The title is not very clear and does not reflect the aim (The aim of this work was to evaluate metabolic imaging using 13C-MRS (Magnetic Resonance Spectroscopy) to assess response to BRAF/MEK inhibition in a syngeneic melanoma model) or the main results of the work (combined HP 13C pyruvate and 13C-glucose fluxomic as a potential marker of response to BRAF/MEK inhibition in melanoma).

Why was studied the 13C-alanine profile shown in 3B? Was it to show that a lactate drop caused by BRAFi/MEKi was not being funneled into the synthesis of alanine from pyruvate? An explanation should be provided.

Some inconsistencies have to be corrected, namely the use of capital letters in the name of drugs, amino acids, etc.; dacarbazine (not dacarbazine),  anova,...

References 51 and 52 should be ordered properly.

Author Response

We would like to thank you for your feedback and suggestions.

It is correct that we did not point out that MCT1 is surprisingly increased in the MEKi group. This is not in accordance with the literature. We noticed that the standard deviation is quite high in this group (higher than in the other groups). This should be confirmed on a larger sample size.

This is now stated in the results section as follows: “ Of note, MCT1 staining was surprinsigly increased in response to MEK inhibition only, yet with a large standard deviation in this group, which remains so far unexplained ».

This point of discussion was also added in the manuscript as follows:

“We did not observe any significant effect of BRAF and/or MEK inhibition, in accordance with the HP pyruvate to lactate exchange data, except for the MCT1 expression that was surprisingly increased in the MEKi treated group, yet with a huge standard deviation in this group. This should be further explored on a larger sample size.”

We adapted the title as suggested, in order to reflect the aim and results of the work.

“Combined HP 13C pyruvate and 13C-glucose fluxomic as a potential marker of response to targeted therapies in YUMM1.7 melanoma xenografts »

Regarding alanine, we studied 13C- alanine since it is the second detectable and quantifiable metabolite issued from 13C-glucose feeding, ex vivo. Here we did not observe any change in 13C-alanine in reponse to BRAF/MEK inhibition.

This point is now added in the discussion as follows: “Of note, we did not observe any significant change in the second quantified ex vivo metabolite, 13C alanine, which is another product of glucose metabolism. If changed it would suggest a major role of aminotransferase enzyme (ALT) in the generation of alanine from pyruvate. »

References and inconsistencies have been corrected in the word version.
